# Clinical Characteristics and Outcome of Patients with Suspected COVID-19 in Emergency Department (RESILIENCY Study II)

**DOI:** 10.3390/diagnostics11081368

**Published:** 2021-07-29

**Authors:** Alessandro Russo, Elio Gentilini Cacciola, Cristian Borrazzo, Valeria Filippi, Tommaso Bucci, Francesco Vullo, Luigi Celani, Erica Binetti, Luigi Battistini, Giancarlo Ceccarelli, Maria Alessandroni, Gioacchino Galardo, Claudio Maria Mastroianni, Gabriella d’Ettorre

**Affiliations:** 1Department of Medical and Surgical Sciences, Infectious and Tropical Disease Unit, “Magna Graecia” University of Catanzaro, 88100 Catanzaro, Italy; 2Policlinico “Umberto I”, Department of Public Health and Infectious Diseases, “Sapienza” University of Rome, 00185 Rome, Italy; Gentilini.1979701@studenti.uniroma1.it (E.G.C.); cristian.borrazzo@uniroma1.it (C.B.); valeria.filippi@uniroma1.it (V.F.); luigi.celani@uniroma1.it (L.C.); erica.binetti@uniroma1.it (E.B.); battistini.1586873@studenti.uniroma1.it (L.B.); giancarlo.ceccarelli@uniroma1.it (G.C.); claudio.mastroianni@uniroma1.it (C.M.M.); gabriella.dettorre@uniroma1.it (G.d.); 3Department of General Surgery, Surgical Specialties and Organ Transplantation “Paride Stefanini”, Sapienza University of Rome, 00185 Rome, Italy; tommaso.bucci@uniroma1.it; 4Department of Radiological, Oncological and Pathological Sciences, “Sapienza” University of Rome, 00185 Rome, Italy; f.vullo@policlinicoumberto1.it; 5Medical Emergency Unit, Sapienza University of Rome, Policlinico Umberto I, 00185 Rome, Italy; m.alessandroni@policlinicoumberto1.it (M.A.); g.galardo@policlinicoumberto1.it (G.G.)

**Keywords:** SARS-CoV-2, COVID-19, decision tree, acute respiratory failure, fever, mortality

## Abstract

Objectives: COVID-19 may show no peculiar signs and symptoms that may differentiate it from other infective or non-infective etiologies; thus, early recognition and prompt management are crucial to improve survival. The aim of this study was to describe clinical, laboratory, and radiological characteristics and outcomes of hospitalized COVID-19 patients compared to those with other infective or non-infective etiologies. Methods: We performed a prospective study from March 2020 to February 2021. All patients hospitalized for suspected or confirmed COVID-19 were prospectively recruited. All patients were evaluated according to a predefined protocol for diagnosis of suspected SARS-CoV-2 infection. The primary endpoint was evaluation of clinical, laboratory, and radiological characteristics associated or not with COVID-19 etiology at time of hospitalization in an emergency department. Results: A total of 1036 patients were included in the study: 717 (69%) patients with confirmed COVID-19 and 319 (31%) without COVID-19, hospitalized for other causes. The main causes of hospitalization among non-COVID-19 patients were acute heart failure (44%) and bacterial pneumonia (45.8%). Overall, 30-day mortality was 9% among the COVID-19 group and 35% in the non-COVID-19 group. Multivariate analysis showed variables (fever > 3 days, dry cough, acute dyspnea, lymphocytes < 1000 × 10^3^/µL, and ferritin > 250 ng/mL) independently associated with COVID-19 etiology. A decision tree was elaborated to early detect COVID-19 patients in the emergency department. Finally, Kaplan–Meier curves on 30-day survival in COVID-19 patients during the first wave (March–May 2020, *n* = 289 patients) and the second wave (October–February 2021, *n* = 428 patients) showed differences between the two study periods (*p* = 0.021). Conclusions: Patients with confirmed diagnosis of COVID-19 may show peculiar characteristics at time of hospitalization that could help physicians to distinguish from other infective or non-infective etiologies. Finally, a different 30-day mortality rate was observed during different periods of the pandemic.

## 1. Introduction

Infection caused by severe acute respiratory coronavirus-2 (SARS-CoV-2) in humans was first described in Wuhan, China, in December 2019 [1,2]. As a consequence of the pandemic, the clinical understanding of the disease has deepened. As a matter of fact, an early recognition and prompt management of coronavirus disease-19 (COVID-19) in hospitalized patients is crucial to improve survival, but COVID-19 infection may not have peculiar signs and symptoms that can differentiate this infection from other infective or non-infective causes of acute respiratory failure and/or fever [3].

In the last months, a huge number of experiences have been published, but despite this, still a large amount of information about this serious disease is missing. To date, all patients admitted to emergency departments with acute respiratory failure and/or fever should be considered as a suspected SARS-CoV-2 infection [4,5,6,7]. A previous experience about the first months of pandemic was already reported (RESILIENCY study I) [8]; the 653 patients from RESILIENCY I have been also included in this analysis.

The main objective of this study was to describe clinical, laboratory, and radiological characteristics and outcomes of hospitalized COVID-19 patients compared to those with other infective or non-infective etiologies. We also reported data about 30-day survival in all COVID-19 patients and during the first wave (March–May 2020) and the second wave (October–February 2021) of the pandemic in Italy.

## 2. Materials and Methods

We performed a prospective, multicenter study (RESILIENCY study) from March 2020 to February 2021. During the study period, patients hospitalized for suspected or confirmed COVID-19 infection were prospectively recruited in 1 large teaching hospital in Rome, Italy. Patients with suspected COVID-19 were admitted to the hospital in case of fever and/or hypoxemic respiratory failure (PaO_2_ < 60 mmHg at rest in ambient air) or of exacerbation of underlying diseases or severe symptoms not manageable outside the hospital. Patients aged < 18 years were excluded. Patients were enrolled in 2 different periods: 1st wave (March–May 2020) and 2nd wave (October–February 2021) of pandemic in Italy.

All patients were evaluated in a dedicated emergency department by a dedicated staff of infectious disease specialists that identified patients with SARS-CoV-2 infection as soon as they arrived at the hospital, followed the patients during the hospital stay, and collected all data prospectively without interfering with the therapeutic decisions. The prospective nature of the study was based on the consecutive enrollment of patients. However, during the first wave, all patients with other etiologies (not COVID-19) were not systematically enrolled in this study. A dedicated emergency department that admitted all patients with fever and/or acute respiratory failure started only during the second wave.

### 2.1. Data Collection and Definitions

The following data were collected from the patients’ medical records at the time of suspected COVID-19 diagnosis in the emergency department: demographics, underlying diseases, clinical, laboratory and radiological findings, treatments, COVID-19-related complications, therapies (including antibiotics, steroids, low-molecular-weight heparin (LMWH), and remdesivir use), and outcome. All complete data were retrieved from an online database for anonymous and automatic data collection.

A confirmed case of COVID-19 was defined by a positive result of reverse transcriptase polymerase chain reaction (RT-PCR) assay of a respiratory sample.

Non-invasive respiratory support techniques included high flow nasal cannulae (HFNC), continuous positive airway pressure, and non-invasive positive pressure ventilation.

### 2.2. Main Outcome Measures

The primary outcome measures included demographics, underlying diseases, clinical presentation of patients affected or not by COVID-19, treatments, intensive care unit (ICU) admission, and all-cause 30-day mortality.

### 2.3. Microbiology

SARS-CoV-2 was identified through real-time (RT)-PCR performed on nasopharyngeal swabs. Respiratory samples were tested using RT-PCR targeted at open reading frame 1ab (ORF 1ab) and nucleocapsid protein (N) genes. A cycle threshold (Ct) value < 37 defined a positive test, while a Ct value ≥ 40 defined a negative result.

Possible co-infection with influenza or other respiratory viruses was ruled out by means of multiplex RT-PCR on the same respiratory sample (Allplex TM Respiratory Panel Assay, Seoul, South Korea).

Bacterial and/or fungal cultures to identify such pathogens were collected according to physicians’ judgment, and micro-organisms were identified with matrix-assisted laser desorption/ionization–time of flight (MALDI-TOF) and tested for antimicrobial susceptibility with Vitek 2 automated system (bioMérieux, Marcy l’Etoile, France).

### 2.4. Statistical Analysis

Primary endpoint was evaluation of risk factors associated or not with COVID-19 infection at the time of hospitalization; secondary endpoint was the evaluation of variables associated with outcome (death or survival) at 30 days.

We developed a decision tree-based predicting model [9,10]. The algorithm has been coded in Python 3.5. As splitting criteria, we used entropy. In order to assess the quality of the model, we performed a 10-fold cross-validation test, considering the sensitivity and specificity as performance measures. Precisely, the algorithm processes the database containing all patients with suspected COVID-19 infection through a progressive sequence of tests. These tests, with a positive or negative outcome for the categorical variable and greater or smaller than a given threshold for the continuous variables, were performed by the algorithm on all the instances in the database. Subsequently, we asked the algorithm to provide the variables that split the samples into classes (COVID-19 and non-COVID-19). For our specific cohort, as the first test, the algorithm selected the variable serum ferritin; this variable divides the root (i.e., the set of all patients) into three leaves: a pure leaf, patients with ferritin more than 346/µL, consisting of 28 non-COVID-19 and 144 COVID-19 patients; with ferritin between 209/µL and 346/µL, consisting of 22 non-COVID-19 and 225 COVID-19 patients; and an impure leaf, patients with ferritin less than 209/µL, consisting of 44 non-COVID-19 and 59 COVID-19 patients. Then, we asked the algorithm to choose a new test, namely, the variable that most divided COVID-19 from non-COVID-19 patients among these subjects. The algorithm identified the variable lymphocytes lesser than equal 1000 × 10^3^/µL, presence of fever >3 days and cough. The new cases of suspected COVID-19 infection will be processed with the built model to predict the probability of developing COVID-19 infection as follows: the patient will be tested according to the sequence of questions identified by the tree and eventually lie in a leaf that will determine the class (COVID-19 or non-COVID-19).

Other statistical analyses were performed using SPSS Statistics version 21.0 (IBM Corp., Armonk, NY, USA) or Microsoft Excel (Office 2019). Quantitative variables are presented using the mean and the standard deviation (SD) if they are normally distributed or the median and the interquartile range (IQR: 25–75%) if they follow a non-normal distribution. Normal distribution was assessed using the Kolmogorov–Smirnov test. The differences between groups were assessed with the chi-squared test or Fisher’s exact test (for categorical variables) and the two-tailed Student’s *t*-test or Mann–Whitney test (for continuous variables), as appropriate. Missing data for each variable were excluded from the denominator. Univariable analysis was used to identify risk factors for COVID-19 or non-COVID-19 etiology and predictors of all cause 30-day mortality. Baseline predictors possibly associated with the outcome at univariable comparison were considered for multivariable logistic regression to estimate adjusted odds ratios (ORs) and 95% confidence intervals (95% CI) for the risk factors including the confounding factors (i.e., age, gender). A stepwise backward selection approach was used to select the predictors to include in the final multivariable model. Survival was analyzed by Kaplan–Meier curves and the statistical significance of the differences between the 2 groups was assessed using the log-rank test.

## 3. Results

Overall, during study period 1036 patients were consecutively included in the study: 717 (69%) patients with confirmed COVID-19 and 319 (31%) without COVID-19, hospitalized for other causes. The main causes of hospitalization, among non-COVID-19 patients, were acute heart failure (44%), bacterial pneumonia (45.8%), pulmonary embolism (6.9%), and other (3.3%). Overall, 136 (18.9%) patients of COVID-19 group and 87 (27.2%) hospitalized for other causes were admitted to intensive care unit.

In Table 1 is reported comparison between patients affected or not by COVID-19 in the emergency department. No difference was reported about male sex, while higher age (75.3 vs. 64.1 years, *p* < 0.001), procalcitonin value (2.6 vs. 0.9 ng/mL, *p* = 0.002), and bacterial co-infection (28% vs. 20%, *p* = 0.004) were more frequently observed in non-COVID-19 group. Conversely, higher mean day from symptoms to RT-PCR test (4 vs. 2, *p* = 0.002), coexisting comorbidities (55% vs. 27%, *p* < 0.001), serum ferritin (458 vs. 343 ng/mL, *p* = 0.013), fever >3 days (77% vs. 12%, *p* < 0.001), and dry cough (47% vs. 16%, *p* < 0.001) were more frequently reported in the COVID-19 group.

Need of oxygen supplement, non-invasive or invasive ventilation, and 30-day mortality are reported in Table 2. No differences were reported about rate of invasive ventilation. Patients in the non-COVID-19 group were more frequently treated with HFNC/NIV (39% vs. 18%, *p* < 0.001), while 30-day mortality was 9% in the COVID-19 group and 35% in the non-COVID-19 group (*p* < 0.001). Mortality rate in COVID-19 group was also calculated using 4C Mortality Score [11]: in patients with 0–3 points, a 30-day mortality of 1.2% was reported, with 4–8 points—12.3%, with 9–14 points—31.7%, and ≥15 points—54.8%.

Multivariable analysis (see Table 3) about risk factors for SARS-CoV-2 etiology showed that fever > 3 days (OR 14, CI95% 9.06–20.07, *p* < 0.001), dry cough (OR 4.06, CI95% 3.03–6.05, *p* < 0.001), acute dyspnea (OR 2.08, CI95% 2.02–3.07, *p* < 0.001), lymphocytes < 1000 ×10^3^/µL (OR 1.05, CI95% 1.01–2, *p* = 0.027), and ferritin > 250 ng/mL (OR 1.05, CI5% 1.02–1.08, *p* = 0.039) were independently associated with COVID-19 at time of hospitalization.

Then, we developed a risk predictive model of COVID-19 using a machine learning technique: decision-making tree to early detect COVID-19 patients in the emergency department. The predictive model highlighted the following variables: serum ferritin, lymphocytes count < 1000/µL, fever > 3 days, and dry cough (see Figure 1). Overall, the sensitivity reached 0.990 (CI95%: 0.964–100), the specificity reached 0.863 (CI95%: 0.737–0.943), and the total accuracy reached 0.954 (CI95%: 0.907–0.981).

Finally, multivariable analysis about risk factors associated with 30-day mortality in COVID-19 patients is reported in Table 4. Age ≥ 65 years (OR 4.23, CI95% 2.83–6.33, *p* < 0.001) and ICU admission (OR 2.51, CI95% 1.44–4.4, *p* = 0.001) were independently associated with 30-day mortality; conversely, no comorbidities (OR 0.03, CI95% 0.02–0.04, *p* < 0.001), steroids (OR 0.16, CI95% 0.1–0.25, *p* < 0.001), LMWH (OR 0.2, CI95% 0.12–0.32, *p* < 0.001), or remdesivir (OR 0.26, CI95% 0.15–0.43, *p* < 0.001) were associated with 30-day survival.

In Figure 2, Kaplan–Meier curves on 30-day survival in COVID-19 patients during the first wave (March–May 2020, *n* = 289 patients) and the second wave (October–February 2021, *n* = 428 patients) showed statistically significant differences between the two study periods (log-rank test *p* = 0.021).

## 4. Discussion

Data from this prospective study highlighted peculiar characteristics associated with SARS-CoV-2 etiology in suspected COVID-19 patients at the emergency department. The findings of the present study, conducted in a large hospital in Italy, showed that the prompt identification of specific clinical characteristics (such as acute dyspnea, dry cough, and duration of fever >3 days) and laboratory findings (such as lymphocytopenia and serum ferritin) can help physicians to distinguish between COVID-19 or other etiologies. As previously reported, the application of a standard approach to management of patients with acute respiratory failure and/or fever [8] associated with the knowledge of clinical and laboratory characteristics of COVID-19 (see Figure 2) can early drive physicians to therapeutic choices. Of importance, our data confirmed that age ≥ 65 years and ICU admission showed an independent association with all-cause 30-day mortality, confirming previous observations [3,4]. Finally, the use of steroids, LMWH, and remdesivir was associated with survival [12,13,14].

Our study has several limitations. First, it was an observational analysis, and we might have missed predictors that could potentially influence the clinical outcome of the patients. Second, this study was performed in a single geographical area of Italy, and the results may not be necessarily representative of other centers. Third, some data were missing for certain patients included in this study; moreover, from November 2020, our institution established a dedicated emergency department that took care of all patients with suspected COVID-19; then, in the previous months (first wave), the patients with other etiologies (not COVID-19) were not systematically enrolled. The compared groups are heterogenous considering different epidemiologic dynamics between the first and second waves. Studies have shown that in the first wave, mostly older people died, while patients in the second wave were younger and the duration of hospitalization and case fatality rate were lower than those in the first wave [15,16]. Finally, no definitive data about the impact of therapies on mortality in COVID-19 patients can be extracted from this analysis.

Data from a large cohort of patients in Moscow [17] showed that typically fever, fatigue, cough, and shortness of breath were the most frequent symptoms observed in suspected COVID-19 patients, in agreement with other studies in other countries [18,19]. Moreover, lymphocytopenia has been previously reported to be associated with higher severity and mortality in COVID-19 patients [20]. Of interest, data from an Italian cohort demonstrated that ferritin levels over the 25th percentile are associated with a more severe pulmonary involvement, independently of age and gender [21]. An early risk assessment for COVID-19 patients from the emergency department, using machine learning, showed that age and measures of oxygenation status were primary indicators of poor patient outcomes [22].

In our cohort, CT findings in COVID-19 patients were frequently indistinguishable from other etiologies, such as acute heart failure, bacterial pneumonia, or pulmonary embolism. Considering typical radiological findings in COVID-19, it is important to underline that in this population, especially in the first stage, could be present only extrapulmonary manifestations with a normal CT finding [23]. In this scenario, it is important to underline that a negative RT-PCR test result does not exclude the possibility of COVID-19, and repeated testing and sampling was shown to improve the sensitivity of RT-PCR [24]. A decision tree classifier for a pre-screening of patients and to conduct triage and fast-track decision, before RT-PCR, were proposed using chest X-ray radiography [25]. Other approaches for diagnosis and to estimate the risk of mortality were reported in order to prioritize medical care and resources [26,27]. Then, a rigorous application of a standard approach to management of suspected COVID-19, associated with clinical and laboratoristic findings as reported in our decision tree, could help physicians in emergency departments.

Of importance, data about 30-day mortality could explain the difference rate of survival (*p* = 0.021) during the first wave (March–May 2020) and the second wave (October–February 2021). A peculiar aspect of COVID-19 patients, especially during the second wave, was the widely use of steroids, LMWH, and remdesivir that could have modified the outcomes of this population [28]. Data in literature showed that the use of dexamethasone resulted in lower 28-day mortality, especially in patients receiving invasive mechanical ventilation [29,30], while important observations were reported about the role of LMWH in survival of COVID-19 patients [31,32]. Of interest, we also observed a different rate of thromboembolic events during the first wave (34.9%) and the second wave (15.7%). Finally, different data were reported worldwide regarding the efficacy of remdesivir, taking into account different outcomes [33,34]. Moreover, our analysis was not performed to weigh all the possible therapeutic confounders: most patients were hospitalized with an extramurally positive test, and a low percentage of patients, with clinical stability, were discharged with a positive test and transferred to “COVID hotels”.

## 5. Conclusions

In conclusion, COVID-19 syndrome is characterized by a heterogeneous clinical, laboratoristic, and radiological presentation, especially at its onset. However, our data showed that patients with a confirmed diagnosis of COVID-19 can show peculiar characteristics at the time of hospitalization that could help physicians to distinguish from other infective or non-infective etiologies [35,36]. Considering the application of a standard approach to management of patients with acute respiratory failure and/or fever, the knowledge of clinical and laboratory characteristics of COVID-19 can drive therapeutic choices early on [37]. We developed a decision tree to detect peculiar characteristics associated with SARS-CoV-2 etiology in suspected COVID-19 patients at an emergency department. However, further larger studies are mandatory to confirm these observations and to validate the decision tree.

## Figures and Tables

**Figure 1 diagnostics-11-01368-f001:**
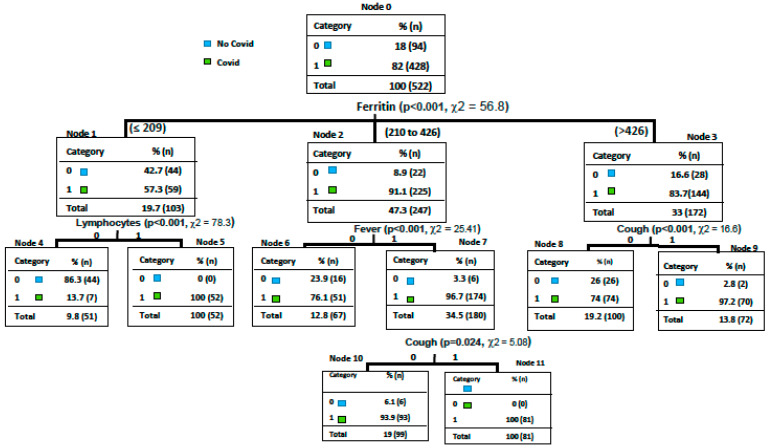
Decision tree to early detect suspected COVID-19 patients in the emergency department.

**Figure 2 diagnostics-11-01368-f002:**
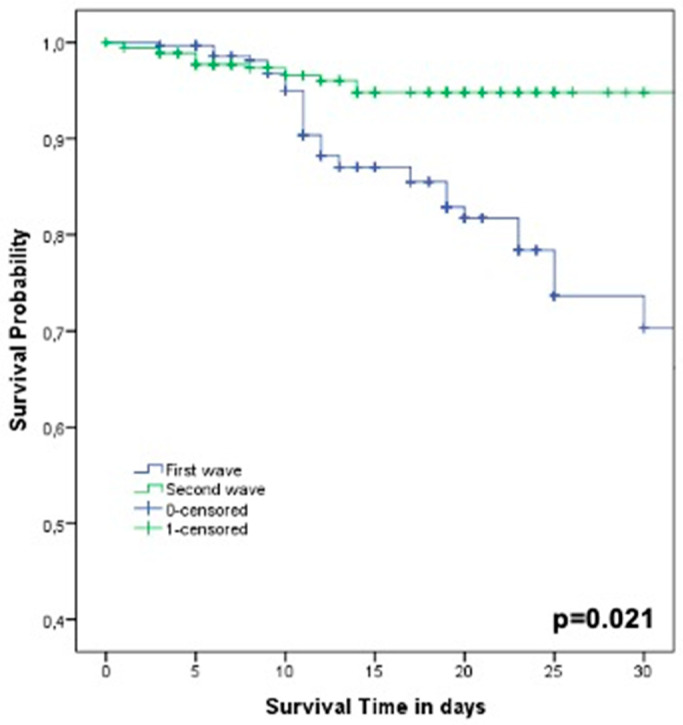
Kaplan–Meier curves on 30-day survival in COVID-19 patients during the first wave (March–May 2020, *n* = 289 patients) and the second wave (October–February 2021, *n* = 428 patients).

**Table 1 diagnostics-11-01368-t001:** Comparison between COVID-19-positive and -negative patients in the emergency department.

Variables	Non-COVID-19 *n* = 319	COVID-19 *n* = 717	*p*-Value
Male sex	199 (62%)	446 (62%)	0.956
Age (years), median (IQR: 25–75%) ± SD	75.3 ± 14.4	64.1 ± 17.1	<0.001
Days from symptoms to RT-PCR test, median (IQR: 25–75%)	2 (1–5)	4 (1–7)	0.002
Coexisting comorbidities, *n* (%)	286 (89.6%)	206 (28.7%)	<0.001
Cardiovascular disease, *n* (%)	248 (78%)	57 (8%)	<0.001
COPD, *n* (%)	163 (51%)	60 (8%)	<0.001
Chronic renal disease, *n* (%)	100 (31%)	37 (5%)	<0.001
Cirrhosis, *n* (%)	64 (20%)	17 (2%)	<0.001
Diabetes, *n* (%)	27 (8%)	77 (11%)	<0.001
Solid lung cancer, *n* (%)	61 (19%)	6 (1%)	<0.001
*Clinical features and radiological findings on admission*			
Fever > 3 days, *n* (%)	39 (12%)	554 (77%)	<0.001
Dry cough, *n* (%)	51 (16%)	334 (47%)	<0.001
Acute dyspnea, *n* (%)	90 (28%)	372 (52%)	<0.002
Gastrointestinal symptoms (diarrhea, abdominal discomfort, nausea, vomiting)	44 (14%)	107 (15%)	0.675
Fatigue, *n* (%)	189 (59%)	109 (15%)	<0.001
Pharyngodynia, *n* (%)	16 (5%)	38 (5%)	1.000
Rhinitis, *n* (%)	184 (58%)	259 (36%)	<0.001
Arthralgia/myalgia, *n* (%)	16 (5%)	73 (10%)	0.007
Anosmia, *n* (%)	7 (2%)	31 (4%)	0.101
Conjunctivitis, *n* (%)	0 (0%)	4 (1%)	0.073
Chest pain, *n* (%)	18 (6%)	33 (5%)	0.508
Signs of overload (limb edema and/or pulmonary stasis), *n* (%)	37 (12%)	17 (2%)	<0.001
Parenchymal thickening, *n* (%)	66 (21%)	344 (48%)	<0.001
Interstitial lung disease, *n* (%)	16 (5%)	31 (4%)	0.464
Pleural effusion, *n* (%)	110 (34%)	191 (27%)	0.022
Cardiomegaly, *n* (%)	99 (31%)	232 (32%)	0.750
Bronchiectasis/emphysema, *n* (%)	27 (24%)	50 (15%)	0.013
*Laboratory findings*			
WBC (×10^3^/µL), median (IQR: 25–75%)	7.5 (6.4–12.3)	6.1 (4.5–8.8)	<0.001
Neutrophils ×10^3^/µL, median (IQR: 25–75%)	5.6 (3.8–9.4)	4.4 (3–7.1)	0.368
Lymphocytes ×10^3^/µL, median (IQR: 25–75%)	1.1 (0.8–1.8)	0.9 (0.6–1.2)	<0.001
Platelets ×10^3^/µL, median (IQR: 25–75%)	232 (192–305)	205 (159–266)	<0.001
D-dimer ng/mL, median (IQR: 25–75%)	820 (367–1322)	631 (340–1242)	0.178
Serum ferritin ng/mL, median (IQR: 25–75%)	343 (120–811)	458 (219–813)	0.013
Procalcitonin ng/mL, median (IQR: 25–75%)	2.6 (0.4–3)	0.9 (0.1–1.5)	0.002
LDH mU/mL, mean ± SD	404 ± 191	317 ± 147	<0.001
CPK U/L, median (IQR: 25–75%)	74 (48–170)	89 (52–160)	0.010
Lactate mmol/L, mean ± SD	1.8 ± 1.4	1.3 ± 0.8	<0.001
C-reactive protein mg/dL, median (IQR: 25–75%)	10.3 (6.5–13)	4 (1.3–9.8)	0.022
Alanine aminotransferase U/L, median (IQR: 25–75%)	28 (20–46)	29 (20–42)	0.258
Aspartate aminotransferase U/L, median (IQR: 25–75%)	26 (18–45)	22 (16–34)	0.015
PaO_2_/FiO_2_, mean ± SD	326 ± 106	317 ± 114	0.010
Bacterial co-infection, *n* (%)	89 (28%)	144 (20%)	0.004
Days of hospitalization, median (IQR: 25–75%)	12 (9–19)	12 (8–19)	0.376
Days to RT-PCR negative test, median (IQR: 25–75%)	-	14 (11–23)	-

SD: standard deviation; COPD: chronic obstructive pulmonary disease; WBC: white blood cells; LDH: lactate dehydrogenase; CPK: creatine phosphokinase.

**Table 2 diagnostics-11-01368-t002:** Need of oxygen supplement, non-invasive or invasive ventilation, and 30-day mortality.

Variables	Non-COVID-19 *n* = 319	COVID-19 *n* = 717	*p*-Value
Invasive ventilation, *n* (%)	27 (8%)	36 (5%)	0.059
Low oxygen flow or room air, *n* (%)	167 (52%)	480 (67%)	<0.001
HFNC/NIV, *n* (%)	126 (39%)	127 (18%)	<0.001
30-day mortality, *n* (%)	111 (35%)	61 (9%)	<0.001

HFNC: high flow nasal cannula; NIV: non-invasive ventilation.

**Table 3 diagnostics-11-01368-t003:** Multivariate analysis about risk factors associated with SARS-CoV-2 etiology.

Variables	OR	CI95%	*p*-Value
Fever > 3 days	14	9.06–20.07	<0.001
Dry cough	4.06	3.03–6.05	<0.001
Acute dyspnea	2.08	2.02–3.07	<0.001
Lymphocytes < 1000 × 10^3^/µL	1.05	1.01–2	0.027
Ferritin > 250 ng/mL	1.05	1.02–1.08	0.039

**Table 4 diagnostics-11-01368-t004:** Multivariate analysis about risk factors associated with 30-day mortality in COVID-19 patients.

Variables	OR	CI95%	*p*-Value
Age ≥ 65 years	4.23	2.83–6.33	<0.001
No comorbidities	0.03	0.02–0.04	<0.001
Steroids	0.16	0.1–0.25	<0.001
LMWH	0.2	0.12–0.32	<0.001
Remdesivir	0.26	0.15–0.43	<0.001
ICU admission	2.51	1.44–4.4	0.001

LMWH: low-molecular-weight heparin; ICU: intensive care unit.

## Data Availability

Data will be available on request by email to alessandro.russo1982@gmail.com.

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
