# Peer review of "Clinical Characteristics and Outcome of Patients with Suspected COVID-19 in Emergency Department (RESILIENCY Study II)"

_diagnostics, 2021, doi:10.3390/diagnostics11081368_

Round 1

Reviewer 1 Report

The authors present a prospective study in which they evaluated factors which may be predictive of COVID-19 infection. They evaluated 1036 patients from a single center in Italy, of which 69% were COVID-19 positive. They report the outcomes and 30-day mortality as well as prediction models for both the diagnosis of COVID-19 and 30-day mortality. 

Major comments:

1) I am surprised that in a large teaching hospital in Rome, in the course of 1 year, only 1036 patients were suspected of having COVID. In many institutions, presenting with shortness of breath would flag the patient for a possible COVID swab and yet only 319 patients with heart failure or pneumonia were admitted in that year. The authors should clarify and provide additional details re: the recruitment of patients into this study as I am concerned there may be significant selection bias.

2) For the 30-day multivariable mortality model, I am unclear what other variables were included in the model that were not significant. Based on the methods, the authors state that only univariable models with p<0.10 would be included and automated backward selection was performed. There are clinically relevant variables such as age and sex that should be included in the model regardless. 

3) The authors state they analyzed survival using KM curves but they report a p-value of 0.021 which is likely a Cox regression analysis? Further clarification is needed. 

4) I am unclear what the main hypothesis/purpose of this article is. If the primary intention relates to predicting COVID-19 diagnosis, I do not think the inclusion of data regarding 30-day mortality or describing survival differences in the 1st and 2nd wave of COVID is relevant to the discussion. I am not opposed to the authors predicting the diagnosis and the 30-day mortality but I agree that predicting mortality is quite challenging given the rapidly evolving therapies/treatments that were present during the second wave which were unknown in the first wave. 

Minor comments:

1) Replace "multivariate" with "multivariable" where applicable 

2) In the introduction, "...COVID-19 infection may have not peculiar signs..." should be "may not have" and "laboratoristic" should be replaced with "laboratory"

3) The data presented in Figure 1 would be much easier to interpret in a table format. 

4) In table 1, consider presenting non-normally distributed continuous variables as median and interquartile range as opposed to mean and SD. 

Author Response

Reviewer 1

The authors present a prospective study in which they evaluated factors which may be predictive of COVID-19 infection. They evaluated 1036 patients from a single center in Italy, of which 69% were COVID-19 positive. They report the outcomes and 30-day mortality as well as prediction models for both the diagnosis of COVID-19 and 30-day mortality. 

Major comments:

1)I am surprised that in a large teaching hospital in Rome, in the course of 1 year, only 1036 patients were suspected of having COVID. In many institutions, presenting with shortness of breath would flag the patient for a possible COVID swab and yet only 319 patients with heart failure or pneumonia were admitted in that year. The authors should clarify and provide additional details re: the recruitment of patients into this study as I am concerned there may be significant selection bias.

R: Dear reviewer, we are grateful for all these important observations. About this suggestion, we want to specify that number of hospitalized patients was in continuous evolution during pandemic. From November 2020 started in our Institution a dedicated Emergency Department in which were admitted all patients with fever and/or acute respiratory failure. In the previous months all patients with other etiologies were not systematically enrolled but, primarily, from June to October no COVID-19 patients were hospitalized. We think that your observation is very important, then we report now in limitations this bias. We specify it also in Methods section.

2) For the 30-day multivariable mortality model, I am unclear what other variables were included in the model that were not significant. Based on the methods, the authors state that only univariable models with p<0.10 would be included and automated backward selection was performed. There are clinically relevant variables such as age and sex that should be included in the model regardless.

R: We agree. The sentence in Methods regarding the purpose of multivariate was reformulated and suggested corrections have been done. 

3) The authors state they analyzed survival using KM curves but they report a p-value of 0.021 which is likely a Cox regression analysis? Further clarification is needed.

R: Comparison of 2 survival curves can be done using a statistical hypothesis test called the log rank test. It is used to test the null hypothesis that there is no difference between the population survival curves (i.e. the probability of an event occurring at any time point is the same for each population). We report now these informations in Methods and Figure 2.

4) I am unclear what the main hypothesis/purpose of this article is. If the primary intention relates to predicting COVID-19 diagnosis, I do not think the inclusion of data regarding 30-day mortality or describing survival differences in the 1st and 2nd wave of COVID is relevant to the discussion. I am not opposed to the authors predicting the diagnosis and the 30-day mortality but I agree that predicting mortality is quite challenging given the rapidly evolving therapies/treatments that were present during the second wave which were unknown in the first wave. 

R: we modified Discussion as required; however, we already discussed the potential differences about mortality in the 2 waves considering the different and evolving therapeutic approaches. Considering also comments of reviewer 2, we decided to modify Discussion section according with all suggestions. We hope that you could understand this strategy about these important observations.

Minor comments:

  • Replace "multivariate" with "multivariable" where applicable 

R: we corrected it.

  • In the introduction, "...COVID-19 infection may have not peculiar signs..." should be "may not have" and "laboratoristic" should be replaced with "laboratory"

R: we corrected it.

3) The data presented in Figure 1 would be much easier to interpret in a table format. 

R: We decided to remove Figure 1.

4) In table 1, consider presenting non-normally distributed continuous variables as median and interquartile range as opposed to mean and SD. 

R: We followed the suggestion and modified the means with standard deviations (SD), which defined the non-normality values, considering them as median and interquartile range (IQR: 25% -75%). Consequently, we modified Table 1 for continuous variables.

Reviewer 2 Report

REVIEW REPORT

Title: Clinical characteristics and outcome of patients with suspected COVID-19 in Emergency Department (RESILIENCY study II)

Article type: Original article

NO: diagnostics-1260932

Date: 23.06.2021

Comments:

The authors (Russo et al) present in their article ((Clinical characteristics and outcome of patients with suspected COVID-19 in Emergency Department (RESILIENCY study II)) a study aimed to find characteristics that could help physicians to distinguish between COVID-19 infection or other etiologies.

To begin with, the authors already published an article titled “Comparison Between Hospitalized Patients Affected or Not Affected by Coronavirus Disease 2019” in Clinical Infectious Diseases (15/6/2021). It is good etiquette to state in your current article that part of the cohort was already utilized or that you have previously tackled a specific research question ((e.g., … the findings of the present study can be summarized as follows: (1) Prompt identification of specific clinical characteristics (eg, dry cough or duration of fever >3 days) and laboratory findings (eg, lymphocytopenia, PaO2/FiO2 ratio <250, procalcitonin value >1 ng/ mL, and lactate >2 mmol/L) can help physicians to distinguish rapidly between COVID-19 or other etiologies; Russo et al., 2021)).

The study premise is interesting. I have several comments:

  • Typos (e.g., … describe clinical, laboratoristic, radiological) – please utilize a language checking software.
  • Abstract – the objectives of the study, as described in the abstract, are not clear.
  • The statement that it can be difficult to differentiate between COVID-19 and other infectious diseases is true. However, also non-infective pathological states? Please elaborate on this.
  • Figure 1: Subheadings for the individual groups (radiological, clinical etc.) would be beneficial for additional clarity.
  • Many studies report a higher mean age in COVID-19 positive deceased. What was your experience? How was the age stratification when considering disease severity?
  • It would be interesting to see a subgrouping of your COVID-19 patients, since it is known that the presentation and severity can differ (asymptomatic infection to severe respiratory distress and failure).
  • I do not understand the coexisting comorbidities value. How come there are so many reported patients with CVD, COPD, renal failure etc. in the no-COVID-19 group, yet the overall assessments of coexisting comorbidities speaks in favor of the COVID-19 group (in terms of higher percentages)?
  • Regarding the values for “Days of hospitalization” and “Days to RT-PCR negative test”. Can you please elaborate on how the days of hospitalization are shorter than the days to RT-PCR negative test (in certain outliers almost 2 times)? Was the initial PCR test always taken at your facility or extramurally? Were the patients, even if still tested positive, discharged?
  • Patients with COVID-19 are at an increased risk of VTE. Current documented rates of incidental VTE in hospitalized patients with COVID-19 ranges from 20–69%, despite the use of pharmacological thromboprophylaxis. One of the important characteristics of the disease, which has led to new anticoagulant strategies etc., is mentioned only once. It would be interesting to see a statement regarding thromboembolic events in both cohorts and a consequent comparison.
  • Can you perhaps comment on the statement that has been previously published by Heldt et al.: “lower patient age contributes to an increased probability of receiving mechanical ventilation and critical care in AICU, while coinciding with lower mortality” (https://www.ncbi.nlm.nih.gov/pmc/articles/PMC7892838/).
  • It would be interesting to see your results in risk stratification of patients compared to the 4C mortality score (https://www.ncbi.nlm.nih.gov/pmc/articles/PMC7116472/).
  • There are several studies comparing different aspects of the first and second wave. Was there anything else that was different between the first and second wave, besides mortality?
  • https://www.ncbi.nlm.nih.gov/pmc/articles/PMC7875012/ (First and second waves of coronavirus disease-19: A comparative study in hospitalized patients in Reus, Spain)
  • https://www.ncbi.nlm.nih.gov/pmc/articles/PMC7875012/ (Second versus first wave of COVID-19 deaths: Shifts in age distribution and in nursing home fatalities)
  • https://journals.plos.org/plosone/article?id=10.1371/journal.pone.0240783 (Decreasing median age of COVID-19 cases in the United States—Changing epidemiology or changing surveillance?)

Other studies that may have relevance, but were omitted:

  • https://www.ncbi.nlm.nih.gov/pmc/articles/PMC8137888/
  • https://www.ncbi.nlm.nih.gov/pmc/articles/PMC7697463/
  • https://www.ncbi.nlm.nih.gov/pmc/articles/PMC7371960/
  • https://www.ncbi.nlm.nih.gov/pmc/articles/PMC7426219/

With best regards.

Author Response

Reviewer 2

The authors (Russo et al) present in their article ((Clinical characteristics and outcome of patients with suspected COVID-19 in Emergency Department (RESILIENCY study II)) a study aimed to find characteristics that could help physicians to distinguish between COVID-19 infection or other etiologies.

To begin with, the authors already published an article titled “Comparison Between Hospitalized Patients Affected or Not Affected by Coronavirus Disease 2019” in Clinical Infectious Diseases (15/6/2021). It is good etiquette to state in your current article that part of the cohort was already utilized or that you have previously tackled a specific research question ((e.g., … the findings of the present study can be summarized as follows: (1) Prompt identification of specific clinical characteristics (eg, dry cough or duration of fever >3 days) and laboratory findings (eg, lymphocytopenia, PaO2/FiO2 ratio <250, procalcitonin value >1 ng/ mL, and lactate >2 mmol/L) can help physicians to distinguish rapidly between COVID-19 or other etiologies; Russo et al., 2021)).

R: Dear reviewer, thank you very much for all your observations. We now state in the Introduction that a previous experience (RESILIENCY study I) was already published.

The study premise is interesting. I have several comments:

  • Typos (e.g., … describe clinical, laboratoristic, radiological) – please utilize a language checking software.

R: we modified as required.

  • Abstract – the objectives of the study, as described in the abstract, are not clear.

R: we modified as required.

  • The statement that it can be difficult to differentiate between COVID-19 and other infectious diseases is true. However, also non-infective pathological states? Please elaborate on this.

R: we modified as required.

  • Figure 1: Subheadings for the individual groups (radiological, clinical etc.) would be beneficial for additional clarity.

R: We decided to remove Figure 1.

  • Many studies report a higher mean age in COVID-19 positive deceased. What was your experience? How was the age stratification when considering disease severity?

It would be interesting to see a subgrouping of your COVID-19 patients, since it is known that the presentation and severity can differ (asymptomatic infection to severe respiratory distress and failure).

R: about these 2 comments we report now in Results section mortality rate according with the 4C mortality score

  • I do not understand the coexisting comorbidities value. How come there are so many reported patients with CVD, COPD, renal failure etc. in the no-COVID-19 group, yet the overall assessments of coexisting comorbidities speaks in favor of the COVID-19 group (in terms of higher percentages)?

R: It is an error. We corrected the Table 1. Thank you very much.

  • Regarding the values for “Days of hospitalization” and “Days to RT-PCR negative test”. Can you please elaborate on how the days of hospitalization are shorter than the days to RT-PCR negative test (in certain outliers almost 2 times)? Was the initial PCR test always taken at your facility or extramurally? Were the patients, even if still tested positive, discharged?

R: about this observation, we want to specify that most of patients were hospitalized with an extramurally positive test. Moreover, a low percentage of patients, with clinical stability, were discharged with a positive test and transferred to “COVID Hotels”.

  • Patients with COVID-19 are at an increased risk of VTE. Current documented rates of incidental VTE in hospitalized patients with COVID-19 ranges from 20–69%, despite the use of pharmacological thromboprophylaxis. One of the important characteristics of the disease, which has led to new anticoagulant strategies etc., is mentioned only once. It would be interesting to see a statement regarding thromboembolic events in both cohorts and a consequent comparison.

R: this is an important observation. We modified Discussion as required.

  • Can you perhaps comment on the statement that has been previously published by Heldt et al.: “lower patient age contributes to an increased probability of receiving mechanical ventilation and critical care in AICU, while coinciding with lower mortality” (https://www.ncbi.nlm.nih.gov/pmc/articles/PMC7892838/).

It would be interesting to see your results in risk stratification of patients compared to the 4C mortality score (https://www.ncbi.nlm.nih.gov/pmc/articles/PMC7116472/).

There are several studies comparing different aspects of the first and second wave. Was there anything else that was different between the first and second wave, besides mortality?

https://www.ncbi.nlm.nih.gov/pmc/articles/PMC7875012/ (First and second waves of coronavirus disease-19: A comparative study in hospitalized patients in Reus, Spain)

https://www.ncbi.nlm.nih.gov/pmc/articles/PMC7875012/ (Second versus first wave of COVID-19 deaths: Shifts in age distribution and in nursing home fatalities)

https://journals.plos.org/plosone/article?id=10.1371/journal.pone.0240783 (Decreasing median age of COVID-19 cases in the United States—Changing epidemiology or changing surveillance?)

Other studies that may have relevance, but were omitted:

https://www.ncbi.nlm.nih.gov/pmc/articles/PMC8137888/

https://www.ncbi.nlm.nih.gov/pmc/articles/PMC7697463/

https://www.ncbi.nlm.nih.gov/pmc/articles/PMC7371960/

https://www.ncbi.nlm.nih.gov/pmc/articles/PMC7426219/

R: about these last observations, considering also comments of reviewer 1, we decided to modify Discussion including only the most relevant articles that you suggested. We report now in Results section mortality rate according with the 4C mortality score However, we want to underline that we decided to report data about mortality in different waves, but a deep analysis is out of the scope of this manuscript. In literature, as you reported, were already published significant studies about comparison between 1 and 2 waves. We hope that you could understand this strategy about your important observations.

Round 2

Reviewer 1 Report

The authors have addressed all my concerns. 

Author Response

R: we are grateful for all your important observations.

Reviewer 2 Report

REVIEW REPORT

Title: Clinical characteristics and outcome of patients with suspected COVID-19 in Emergency Department (RESILIENCY study II)

Article type: Original article

NO: diagnostics-1260932

Version: second (first revision)

Date: 01.07.2021

Comments:

The authors have resubmitted their revised version of the manuscript.

Overall, the previously raised concerns have been mostly addressed adequately. However, there are still some points that need further clarification:

  • Abstract – I believe the aim of the study should be mentioned under “objectives” and not under methods. Further clarification would be appreciated.
  • The authors have added a sentence regarding their previous publication. I would advise the authors to reformulate or simply rethink the sentence order of the paragraph, so readers are not confused about the wording and sequence of thoughts (which applies to which statement and publication). Also, please indicate to what extent the first study group was reused in this article. Perhaps simply a sentence: “The study group of 653 patients from RESILIENCY I have been also included in the calculations.”
  • Line 61 – Please clarify whether it was a continuous interval or 2 separate intervals in which you collected patients. Also, if so, please indicate how many patients were collected in each phase
  • The sentence about patient evaluation (lines 69-70). Were only patients identified who had SARS-CoV-2 pneumonia? I suppose all patients with SARS-CoV-2 infection, or not?
  • The added sentence about the timeframe prior to November 2020 is somewhat oddly phrased. I do not fully understand it (lines 73 – 75).
  • Line 161 – The table title is phrased somewhat oddly … perhaps - Comparison between COVID-19 positive and negative patients.
  • Line 250 – the sentence regarding the dedicated Emergency department is somewhat oddly phrased. Perhaps … in November 2020 our Institution introduced/established a dedicated ED, which took care of all patients with …
  • A citation regrading pulmonary thrombosis or VTE should be added in the discussion section.
  • Regarding your answer “about this observation, we want to specify that most of patients were hospitalized with an extramurally positive test. Moreover, a low percentage of patients, with clinical stability, were discharged with a positive test and transferred to “COVID Hotels”.« - I as a reader would be interested in this information. Perhaps it could be added in the discussion section.

With best regards.

Author Response

The authors have resubmitted their revised version of the manuscript.

Overall, the previously raised concerns have been mostly addressed adequately. However, there are still some points that need further clarification:

  • Abstract – I believe the aim of the study should be mentioned under “objectives” and not under methods. Further clarification would be appreciated.

R: we modified as required.

  • The authors have added a sentence regarding their previous publication. I would advise the authors to reformulate or simply rethink the sentence order of the paragraph, so readers are not confused about the wording and sequence of thoughts (which applies to which statement and publication). Also, please indicate to what extent the first study group was reused in this article. Perhaps simply a sentence: “The study group of 653 patients from RESILIENCY I have been also included in the calculations.”

R: we modified as required.

  • Line 61 – Please clarify whether it was a continuous interval or 2 separate intervals in which you collected patients. Also, if so, please indicate how many patients were collected in each phase

R: we modified as required.

  • The sentence about patient evaluation (lines 69-70). Were only patients identified who had SARS-CoV-2 pneumonia? I suppose all patients with SARS-CoV-2 infection, or not?

R: all hospitalized patients with SARS-CoV-2 infection were included in the study, we corrected the sentence. Thank you very much!

  • The added sentence about the timeframe prior to November 2020 is somewhat oddly phrased. I do not fully understand it (lines 73 – 75).

R: we reformulated it.

  • Line 161 – The table title is phrased somewhat oddly … perhaps - Comparison between COVID-19 positive and negative patients.

R: we modified as required.

  • Line 250 – the sentence regarding the dedicated Emergency department is somewhat oddly phrased. Perhaps … in November 2020 our Institution introduced/established a dedicated ED, which took care of all patients with …

R: we reformulated it

  • A citation regrading pulmonary thrombosis or VTE should be added in the discussion section.

R: we introduced a reference (n°30).

  • Regarding your answer “about this observation, we want to specify that most of patients were hospitalized with an extramurally positive test. Moreover, a low percentage of patients, with clinical stability, were discharged with a positive test and transferred to “COVID Hotels”.« - I as a reader would be interested in this information. Perhaps it could be added in the discussion section.

R: we introduced this sentence in Discussion, as required. Thanks for another important observation.

Round 3

Reviewer 2 Report

REVIEW REPORT

Title: Clinical characteristics and outcome of patients with suspected COVID-19 in Emergency Department (RESILIENCY study II)

Article type: Original article

NO: diagnostics-1260932

Version: second (second revision)

Date: 09.07.2021

Comments:

The authors have resubmitted their second revised version of the manuscript.

Overall, the previously raised concerns have been mostly addressed adequately.

One remaining concern:

  • Your populations are quite heterogenous and the sampling is confusing. To avoid statements about potential selection bias I would advise an additional statement about your limitations. The compared groups are heterogenous in nature due to different epidemiologic dynamics between the first and second wave. Studies have shown, that in the first wave mostly older people died. According to some studies, patients in the second wave were younger and the duration of hospitalization and case fatality rate were lower than those in the first wave (https://journals.plos.org/plosone/article?id=10.1371/journal.pone.0248029; https://www.nature.com/articles/s41591-020-1112-0).
  • The above should be examined by the academic editor.

With best regards.

Author Response

The authors have resubmitted their second revised version of the manuscript.

Overall, the previously raised concerns have been mostly addressed adequately.

One remaining concern:

  • Your populations are quite heterogenous and the sampling is confusing. To avoid statements about potential selection bias I would advise an additional statement about your limitations. The compared groups are heterogenous in nature due to different epidemiologic dynamics between the first and second wave. Studies have shown, that in the first wave mostly older people died. According to some studies, patients in the second wave were younger and the duration of hospitalization and case fatality rate were lower than those in the first wave (https://journals.plos.org/plosone/article?id=10.1371/journal.pone.0248029; https://www.nature.com/articles/s41591-020-1112-0).

The above should be examined by the academic editor.

R: we modified as required Discussion, including these references.
